# Saliva Diagnosis as a Disease Predictor

**DOI:** 10.3390/jcm9020377

**Published:** 2020-01-30

**Authors:** Patrick L McGeer, Moonhee Lee, Krista Kennedy, Edith G McGeer

**Affiliations:** Aurin Biotech INC, 4727 West 2nd Avenue, Vancouver, BC V6T 1C1, Canadamcgeer@mail.ubc.ca (E.G.M.)

**Keywords:** diagnostic methodology, gene expression, Abeta42, C-reactive protein, tumor necrosis factor

## Abstract

Background: Saliva, the most readily available body fluid, is the product of genes which are in constant activity throughout life. Measurement of saliva can predict the onset of some diseases years before their accumulation in vulnerable tissues causes clinical signs to appear. The purpose of this study was is to demonstrate current applications of saliva analysis and to predict and prevent disease progression. Methods: We measured levels of Abeta42, C-reactive proteins (CRPs), and tumornecrosis factors (TNFs) in saliva from both healthy and fatal diseased cases such as cancer, Alzheimer’s disease (AD), and coronary heart disease by ELISA-mediated techniques. We also immunostained human tissue sections with antibodies specific to these proteins to demonstrate the data are comparable. Results: We found all the proteins expressed constantly in saliva from healthy controls but increased in diseased cases. This was accompanied by data from immunohistochemistry. It was also found that these proteins wereexpressed in high amounts in some healthy controls, which reflects high risk for the onset of diseases such as AD and heart diseases.Conclusions: It is concluded that measuring changes in essential gene products in saliva can predict onset of fatal diseases and open the door to effective protection measures, thus preventing premature death.

## 1. Introduction

Saliva is a product of the salivary glands. It is expressed at a constant rate around the clock every day of life as is the case for all proteins generated from maintenance genes. Such genes do not express themselves here and there and now and then, as many have believed. They are expressed everywhere in the body at a rate specific for the tissue involved. Once the relative rates for tissues are known, they will apply to the proteins which are the products of their genetic activity [1].

A suitable example is amyloid beta protein terminating at position 42 (Abeta42). Immunohistochemical studies of brain tissue in Alzheimer’s disease (AD) cases have established intense staining for Abeta42 in affected regions. It was known that Abeta42 accumulation in AD is initiated by Abeta42 binding to the complement protein C1q, which initiate complement cascade resulting in neuronal death in affected area by membrane attack complex formation [2], which eventually results in the cognitive deficits that define clinical AD. In the affected areas, the gene products have not been clearly removed.This means that the differential effect of gene activity results in an integral effect of tissue accumulation. It has been suggested that this sequence of events can be avoided by early intervention with anti-inflammatory agents. Traditional non-steroidal anti-inflammatory drugs (NSAIDs) are one such class of agents. [3]. Therefore, it is useful to investigate whether Abeta42 protein expression is increased in AD patients and whether oral administration of a NSAID such as ibuprofen attenuates its expression.

C-reactive proteins (CRPs) are an acute-phase protein produced in the liver and have been considered as a biomarker with plasma levels increasing in response to inflammation [4]. It was suggested that increased CRP levels were found in cardiovascular diseases such as coronary heart disease [5]. Therefore, a study of CRP levels in saliva and prediction of heart disease onset is a good example of the relationship between gene activity and diseases.

Tumornecrosis factor-α (TNF) is produced as a 233 amino acid transmembrane protein of molecular weight of 26 kDa and is stable in trimers [6]. It is known to be upregulated in inflammatory conditions such as malignant tumor expansion since it results in neo-capillary formation; cancers may be relatively benign until factors, yet to be identified, induce TNF production. This new, excessive production of TNF leads to a cachectic state. The apparent reason is that the proliferating cancer cells need TNFto stimulate their necessary blood supply. This requires new capillary formation. Complement activation is required to produce the necessary vascular endothelial growth factor for this to occur [6,7]. In order to clarify the functions of TNF, measuring salivary TNF levels as an indicator of its in vivo expression in healthy and cancerous groups will be necessary to understand the relationship between TNF gene activity and cancer cell growth.

Urine and blood analyses have been routine for many decades but saliva, another body fluid, has been largely less attended to [8], and yet, it is beginning to appear that saliva may not only offer an index to the existence of a number of diseases, but may even offer a clue to vulnerability to some, where protective action can be taken.

## 2. Materials and Methods

### 2.1. Immunohistochemistry

Elderly brain tissue from control, cancer, and Alzheimer’s disease cases were selected for Abeta42 and TNF immunohistochemistry. Immunohistochemistry was carried out as previously described in detail [9]. Briefly, brain tissues were fixed in 4% paraformaldehyde and after 3–4 days transferred to a 15% buffered sucrose maintenance solution. For single immunostaining, 30 µm sections were pretreated with 0.5% H_2_O_2_, blocked with 5% skim milk, and incubated in the primary antibodies, mouse monoclonal anti-human Abeta42 antibody (1/1000, Cat #: 6F/3D, DAKO, Mississauga, Ontario, Canada) or rabbit polyclonal anti-TNF antibody (1/1000, MyBioSource,Cat #: MBS7002356, San Diego, Canada),in PBS-T overnight at room temperature. Sections were next treated with the appropriate biotinylated secondary antibody (DAKO, Mississauga, Ontario, Canada, 1/2000) for 2 h at room temperature, followed by incubation with the avidin-biotinylated horseradish peroxidase complex (DAKO, Mississauga, Ontario, Canada) for 1 h at room temperature. Peroxidase labeling was visualized by incubation in 0.01% 3,3′-diaminobenzidine containing 1% nickel ammonium sulfate, 5 mM imidazole, and 0.001% H_2_O_2_in 0.05 M Tris-HCl buffer, pH 7.6. When a dark purple color developed, sections were mounted on glass slides. For CRP staining hearts from healthy persons and patients with coronary heart attack were used. A sheep polyclonal anti-CRP antibody (1/1000, Wako Chemicals,Cat #: A073, Richmond, VA, USA) was used for immunostaining. 

### 2.2. Saliva Collection

Saliva (5–10 mL) from normal healthy controls and patients was collected in vials including 100 μL of thioflavin S and sodium azide (0.5 mg/mL each) to prevent amyloid aggregation induced by Abeta42 proteins and bacterial growth. Detailed demographic information of all the saliva donors (both normal controls and all the diseased cases) are provided in Appendix A.

### 2.3. Abeta 42 ELISA Assays 

For determining the Abeta42 levels, an ELISA type test was developed. The first step was adding an Abeta42 capture antibody to 96-microwell plates. This was achieved by adding 100 μL of the rabbit polyclonal anti-Abeta42 antibody from Novus Biochemicals (NBP2-44113, Littleton, CO, USA) diluted 1/1000 in PBS to the plates. They were left overnight at 4 °C to allow the antibody to bind. The supernatants were then discarded. Next, to block non-specific binding to the capture antibody, 400 μL of 5% bovine serum albumin (BSA, Sigma, St. Louis, MO, USA) in PBS-T (PBS with 0.5% Tween-20) was added to the plates, followed by incubation at 20 °C for 1 h. The BSA solution was discarded, and the wells were washed twice with PBS. Abeta42-containing samples at various concentrations (0–100 pg/mL) and human saliva (100 μL) were added to the plates and incubated at room temperature (RT) for 3 h. The supernatants were then discarded. The wells were washed twice with PBS-T. They were next incubated with a monoclonal antibody to Abeta42 (HRP-linked optimAb beta-amyloid 1–16 monoclonal Ab, Eurogentec, Fremont, CA, USA). It was added as 100 μL of a 1/500 dilution in 5% BSA including PBS-T. Incubation was carried out at room temperature for 1.5 h. After the plate was washed twice with PBS-T and once with PBS, tetramethylbenzidine (TMB) solution (100 μL, Thermofisher Scientific, Cat#: N301,) was added and incubation carried out at 37°C for 5 min (TMB solution kept at 4°C until it is added for color reaction). A stop solution of 100 μL (2 M sulfuric acid, Thermofisher Scientific, Cat#: N600, Webpage: https://www.thermofisher.com/order/catalog/product/N600) was added to the well plates. Optical density was measured at 450 nm. The Abeta42protein concentration was calculated from the standard curve which was made with human Abeta42peptides (Anaspec, 0.5 mg, Cat#:AS24224). 

### 2.4. C-Reactive Protein Assay

For determining the C-reactive protein levels (CRPs), the ELISA protocol was utilized. The first step was adding a CRP capture antibody (mouse monoclonal, 1/2000) to 96-microwell plates. The capture antibody was kindly supplied by Dr. Nijmeije (Amsterdam, the Netherlands). The plate was left overnight in a cold room (at 4 °C) to allow binding of the antibody. The remaining supernatants were then discarded. Bovine serum albumin (5%, 400 μL) in PBS-T was added to the plates to block non-specific binding to the capture antibody. The solution was then incubated at 20 °C for 1 h. That BSA solution was discarded, and the wells were washed twice with PBS.

Human saliva (100 μL) samples and CRP at various concentrations (0–50 ng/mL) were added to the plates and incubated at room temperature for 3 h. The saliva supernatants were then discarded. The wells were washed twice with PBS-T and were incubated with rabbit polyclonal anti-CRP antibody (100 μL, 1/2000, Dako, Cat#: A073) at room temperature for 1.5 h. After that, the plate was washed twice with PBS-T HRP-conjugated polyclonal anti rabbit IgG antitibody (100 μL, 1/2000, Dako, Cat#: P0448, DAKO, Mississauga, Ontario, Canada) for 1 h. The antibody was removed, and the wells were washed twice with PBS-T and once with PBS.

For detection, a tetramethylbenzidine (TMB) solution (100 μL, Thermofisher Scientific, Cat#: N301) was added and the incubation carried out at 37 °C for 5 min (the TMB solution was kept at 4 °C until it was added for the color reaction). A stop solution of 100 μL (2 M sulfuric acid, Thermofisher Scientific, Cat#: N600) was then added to the well plates. The optical density was measured at 450 nm. The CRP concentration was calculated from a standard curve which was made with recombinant CRP (BioLegend, San Diego, CA, USA).

### 2.5. Tumor Necrosis Factor (TNF) Assay

Quantitation of TNF was performed with ELISA detection kits (Peprotech, NJ, USA) following protocols described by the manufacturer. Briefly, capture antibody at a concentration of 1 μg/mLwas plated on 96-well plates. The plate was sealed and incubated at 37 °C overnight. After the antibody solution was discarded, the plate wells were washed four times with 300 μL wash buffer. Blocking buffer (300 μL) was added to each well, followed by incubation for 1 h at room temperature. The blocking buffer was discarded, and the plate wells were washed four times with 300 μL wash buffer. Saliva (100 μL) was added to each well and the plate was incubated at RT for 2 h. After washing with buffer, 100 μL detection antibody (0.15 μg/mL) was added to each well and the wells were incubated at RT for 2 h. Then streptavidin-HRP conjugates were incubated at RT for 30 min, followed by a color reaction with 100 μL TMB for 20 min. The TNF concentration was determined from a standard curvemade with the TNF peptide available in the kit.

### 2.6. Data Analysis

The significance of differences between data sets was analyzed by one-way ANOVA. Multiple group comparisons were followed by a post-hoc Bonferroni *t*-test. *p* values are given in figure legends.

## 3. Results

We firstly investigated whether Abeta42 protein accumulation occurs in the brains of patients who died with post-mortem evidence of AD brains. A suitable control brain was available from a person who died suddenly from a heart attack. The brains were stained with monoclonal Abeta42 antibody. Figure 1A and B demonstrate that Abeta42 accumulation in the hippocampal area is clearly evidenced in AD (Figure 1A) but not in the healthy control (Figure 1B). 

Since it is known that there is an Abeta42 increase in cerebrospinal fluid from AD patients, we examined whether there was a similar increase in their saliva. Saliva (5–10 mL) was collected in plastic vials. which also included 100 μL of thioflavin S and sodium azide (0.5 mg/mL each) to prevented not only aggregation induced by Abeta42 protein but also bacterial growth.Abeta42 levels were measured by an ELISA method. Protein amounts were analyzed by a standard curve made with commercial Abeta42 proteins. The data in Figure 2A show that in saliva, the low-level control group expressed protein levels that were remarkably constant between the age of 16–92, with no difference occurring between males and females (Appendix A). This low-level control group expressed Abeta protein levels lowerthan 30 pg/mL. The 148 low-level control group cases, including one case with Parkinson’s disease (diagnosed 3 years previously), showed a mean ± SD of 21.26 ± 1.73 pg/mL. We also found that the89 cases in the high-level control group showed Abeta42 protein levels greater than the 148 cases low-level control group. This high-level control group expressed levels greater than 30 pg/mL. They are believed to be at high risk for AD because their family history showed 1.8 times higher Abeta42 levels than low-level controls (mean ± SD: 37.96 ± 8.13 pg/mL, *p* < 0.01 compared with low-level control group). The ages of this group were mainly distributed between 40 and 80 years (Figure 2A). 

We studied whether administration of ibuprofen (200 mg, twice a day) could reduce their Abeta42 protein levels. For this study, eleven cases were selected in the high healthy control group. They orally administered ibuprofen (200 mg, twice a day) and measured their saliva Abeta42 protein levels every one or two weeks. We found that all the high controls showed their salivary Abeta42 protein levels in the saliva to be reduced to low control levels within a month. Two of them are shown in Figure 2B. 

We also measured salivary Abeta42 protein levels from patients with AD. The 30 AD cases had been diagnosed between 3 and 10 years previously in local clinics and hospitals. They showed Abeta42 protein levels even greater than the high-level control groups (mean ± SD: 51.70 ± 10.50 pg/mL, *p* < 0.05 compared with the high-level control group). As a further control, we measured Abeta42 protein throughout the day and found that there was no change in the protein levels (Figure 2C). Overall, these data demonstrate that measuring salivary Abeta42 levels can diagnose AD and indicate as well that it may predict the risk of future onset.

The next example we investigated was C-reactive protein which is known to be involved in coronary heart disease. We examined the expression of the proteins in heart tissues. We immunostained the heart tissue from patients with coronary heart attack with sheep polyclonal anti-CRP antibody. We found that expression of CRP was increased in heart tissue from patients who died with coronary heart attack (Figure 1D) compared with tissue from a healthy control, which was from a patient who died ina car accident (Figure 1C).

We used theELISA assay method to measure salivary CRP protein levels. Data were analyzed with a standard curve with commercially available recombinant human CRP proteins. Figure 3 demonstrates the diagnosis and predictability of inflammation, especially coronary heart attack, by measurements of salivary CRP levels in 66 individuals. Healthy people show CRP levels of less than 3 ng/mL in their saliva (low-level control group, 31 cases, mean ± SD: 1.67 ± 0.21). As shown for Abeta42, healthy controls (low-level controls) produced CRPs constantly over the ages from 12 to 90 with no gender difference. Diseases such as Parkinson’s disease and type II diabetes did not affect CRP expression (1.83 and 1.84 ng/mL, respectively). 

However, high-level control group cases, who demonstrated CRP levels higher than 3 ng/mL, showed increased levels of CRP with no overlap with thelow-level control group (mean ± SD: 5.34 ± 1.70, *p* < 0.01 compared with low-level control group). This group is believed to be at risk for onset of inflammatory diseases, including heart disease. Demographically, everybody in this group showed a clear family history of cardiovascular disease such as heart attack or hypertension.

The diseased group in both genders showed CRP levels high enough to suffer various diseases such as heart attack, transient ischemic attack (TIA), peripheral nerve failure, and giant cell arthritis (mean ± SD: 15.20 ± 7.21, *p* < 0.01 compared with the high-level control group). 

Another example was tumornecrosis factor-α (TNF). For immunostaining, we used a healthy control brain tissue from a person who died from a car accident. The brain tissue from a diseased case was from a patient who died from leukemia. We immunostained the capillary region of the brains with rabbit polyclonal anti-TNF antibody. The data in Figure 1E,F show immunohistochemical staining for TNF in brain tissue from the healthy case (E) and the cancer case (F). Intensely immunostained capillaries in the cancer case compared to the normal case were found. 

In order to confirm the results, we examined salivary TNF protein levels. It was found that the predictability of cancer onset by measuring salivary TNF levels in 42 people is possible (Figure 4). Salivary TNF levels in 25 healthy volunteers, shown in Appendix A, ranging from 12 to 79 years of age were in the very narrow range of 155–175 pg/mL (control level group, mean ± SD: 159.18 ± 7.38). They are consistent with the presumed role of TNF in aiding capillaries to provide a normal blood supply to tissues. Individuals with various types of cancer (15 cases), diagnosed in local clinics and hospitals 1–3 years previously, showed salivary TNF protein levels which were 2–3 fold higher than healthy control group. The values ranged from 260 to 474 pg/mL (mean ± SD: 351.53 ± 63.8). Presumably, this was because the malignancies demanded neo-capillarization to supply their needs. There was no overlap between the levels in normal and cancer cases (*p* < 0.01 compared with control level group).

## 4. Discussion

Saliva is clearly an excellent material to use for the diagnosis of several of the more important diseases of aging [10] and, as indicated by the results with some people presumably at risk for AD because of family histories, may also indicate the probability and the effect of possible intervention. Of course, it will take many years and a large number of wider studies before these facts are established. There is clear evidence of inflammation in AD brain [11], and a number of epidemiological studies show that chronic use of anti-inflammatories such as ibuprofen markedly lessen the risk [12]. The mechanism for the effect of ibuprofen on the production of abeta42 protein remains to be determined.

An aspect of some interest is that expression rates of these proteins are constant. Samples were collected each 4 h over a 24 h from normal individuals to demonstrate this (Figure 2B). No aging effects were shown (Appendix A).

As a test for the diseases, we measured the saliva levels of Abeta42 for AD, CRP for heart disease, and TNF for cancer. All the biomarkers were expressed at constant levels in healthy controls but were significantly increased in the diseased cases (Figure 2, Figure 3 and Figure 4). This agreed with the immunohistochemical results (Figure 1). Some healthy controls expressed high amounts of Abeta42 or CRP and are presumably at risk for the diseases (Figure 2 and Figure 3). Importantly, we found that Abeta42 levels in high controls were reduced when they took ibuprofen regularly (200 mg, two per day) (11 cases tested and Figure 2B shows two cases). The data shows that reducing inflammation reduces saliva Abeta42 expression. 

There are a few references to the use of salival Abeta42 in diagnosing AD [13,14], of saliva alpha-synuclein as a biomarker for Parkinson’s disease [15] and progressive supranuclear palsy [16], and quite a few on saliva levels of CRP in heart disease and atherosclerosis I [17,18].

There is also a fairly extensive literature, largely from the orient and in dental journals, on the use of analyses for TNF in saliva in diagnosing oral cancer [19,20] but nothing on its broader application to systemic disease. Thus, there is ample evidence that analyses of saliva can be very useful for diagnostic purposes. A review on the diagnostic potential of saliva was published in 2011 [21] but received relatively little attention. The data accumulated since that time and the evidence that the levels of at least some constituents remain constant during the day may promote its further use. 

## Figures and Tables

**Figure 1 jcm-09-00377-f001:**
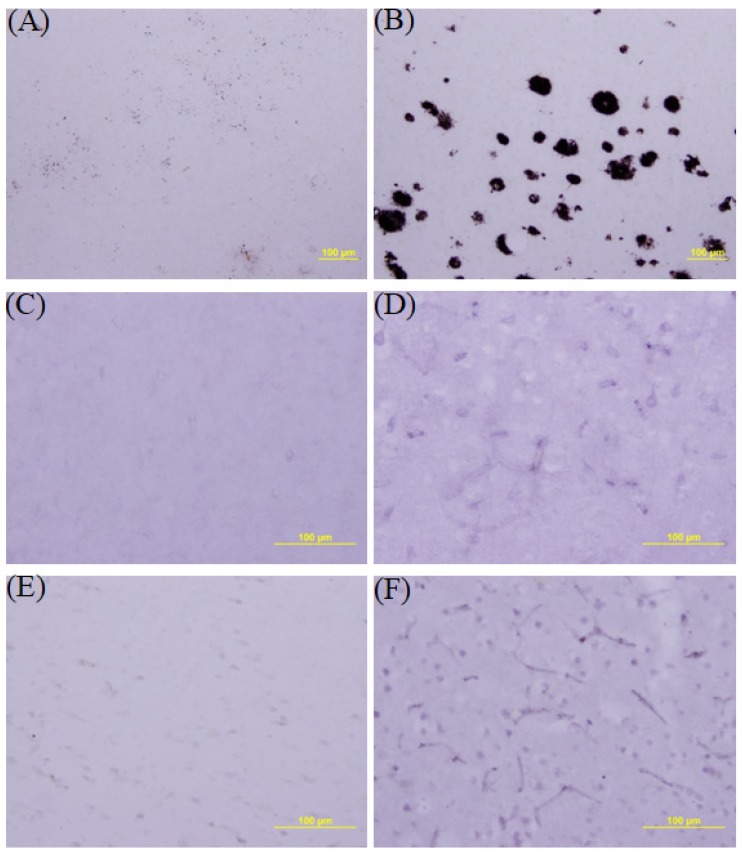
Abeta42 immunostaining of the hippocampal area of brain tissue from a healthy case (**A**) and from a patient dying with Alzheimer’s disease(AD) (**B**). Abeta 42 accumulation was found in the AD brain. C-reactive protein (CRP) immunohistochemical data of hearts from a healthy control case (**C**) and from a patient dying with coronary heart attack (**D**). CRP expression was increased in the coronary heart attack case compared with the healthy case. Immunohistochemical staining oftumornecrosis factor(TNF) in brain tissue from a healthy case (**E**), and a cancer case (**F**). Notice the intense immunostaining of capillaries in the cancer case compared to the normal case.

**Figure 2 jcm-09-00377-f002:**
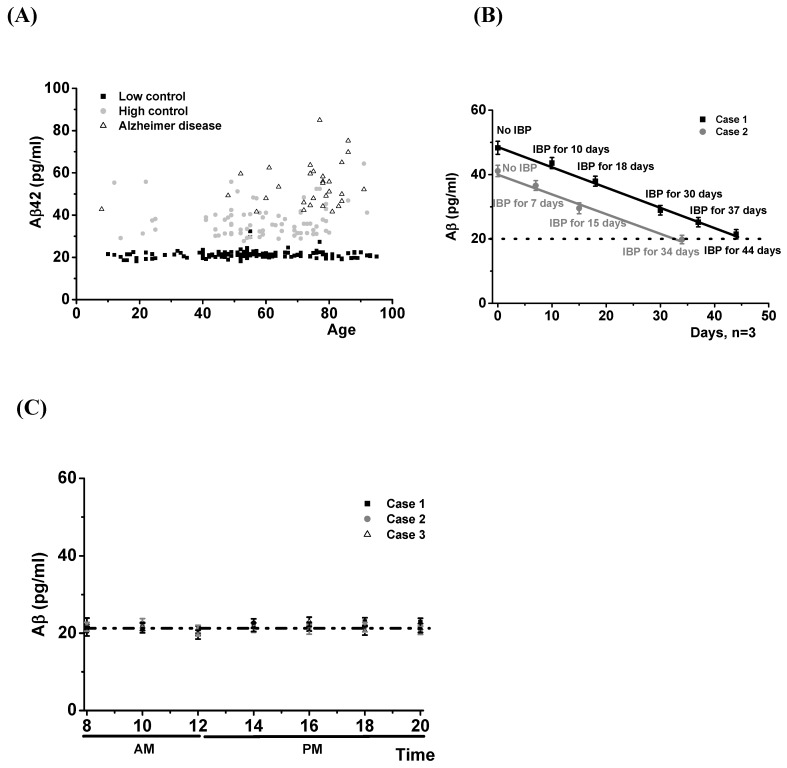
(**A**) Abeta42 levels in the saliva of 148 normal, 89 high normal (at risk), and 30 AD cases. Aβ levels of low controls, high controls, and AD were tested. Averages and SD were 21.26 ± 1.73 for low controls (148 cases), 37.96 ± 8.13 for high controls (89 cases), and 51.70 ± 10.50: *p* < 0.01 between low level control group, high level control group, and AD group and *p* < 0.05between high level control group and AD. (**B**) A reduction in salivary Abeta42 by administration of ibuprofen (dosage: 200 mg, twice a day). Case 1 in black showed 48.31 pg/mL in salivary Abeta42, which was reduced to normal in 44 days. In Case 2, salivary Abeta42 was 41.12 pg/mL. This was reduced in 34 days on ibuprofen. (**C**) Salivary Abeta42 levels were constant during the day. Three cases were examined.

**Figure 3 jcm-09-00377-f003:**
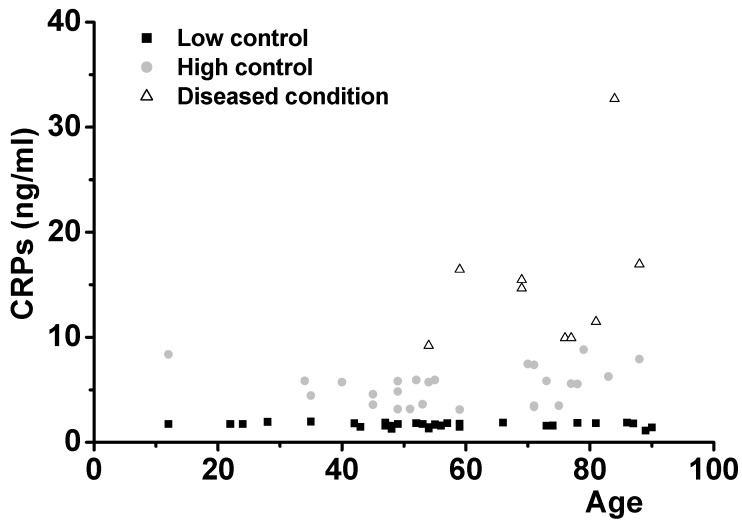
C-reactive proteins (CRPs) levels of low controls, high controls, and diseased conditions were tested. Averages and SD were 1.67 ± 0.21 for the low-level control group (31 cases), 5.34 ± 1.70 for the high-level control group (26 cases), and 15.20 ± 7.21 for the 9 diseased cases. One-way ANOVA was carried out to test the significance of the data: *p* < 0.01 between low controls, high controls, and diseased conditions and *p* < 0.01 between high controls and diseased conditions.

**Figure 4 jcm-09-00377-f004:**
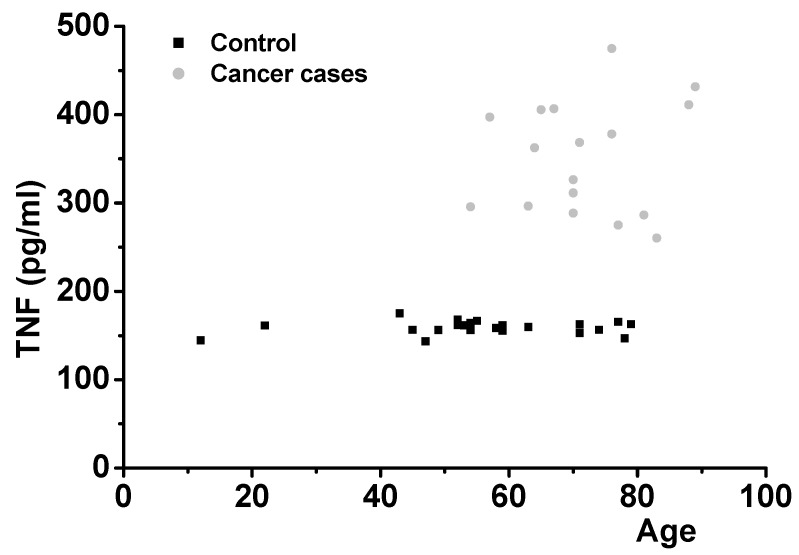
TNF levels in controls and cancer cases were tested. Averages and SD were 159.18 ± 7.38 for controls (25 cases) and 351.53 ± 63.8 for cancer cases (17 cases). The values ranged from 260 pg/mL to 474 pg/mL. There was no overlap between normal and cancer cases. One-way ANOVA was carried out to test the significance of the data: *p* < 0.01 between controls and cancer cases.

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
