# Peer review of "Saliva Diagnosis as a Disease Predictor"

_jcm, 2020, doi:10.3390/jcm9020377_

Round 1

Reviewer 1 Report

In this study, authors measured levels of Aβ42, C-reactive protein (PCR) and TNFα in saliva from healthy subjects and patients with cancer, AD and coronary heart disease. However, no data, including demographic and clinical information, has been reported, thus it is imposible to compare any results, neither to get any conclusion.

Methods should be written more accurately. Demographic and clinical data of subject/patient groups is missing. Additionally, no data regards saliva collection is reported. Procedure of CRP assay is missing (sample used, volumen,...)

Statistical analysis is completely deficience. I recommend a complete revision for the statistical analysis: Use statistical tests accordingly (parametric or non-parametric). If ANOVA is used, which post hoc test is used to determine individual differences? For data visualization please use SD instead of SEM.

Please rewrite Results to highlight the findings of this study. Page 3: "The data in Figure 2A ...expressed levels that were...". Levels of what? Where? In which sample? "...showed a mean of 21.26±0.14pg/ml". Mean of what? Where? These are only a couple of examples in the manuscript, but there are many others. These statements do not support the findings proposed by the authors in this manuscript.

What is "high controls"?

Authors should improved the quality of immunohistochemistry images.

Author Response

We are pleased to resubmit our manuscript "Saliva Diagnosis as a Disease Predictor (Manuscript ID: jcm-681440)" with revisions as suggested by the reviewers. We thank the reviewers for going over the manuscript so carefully and for making constructive suggestions for improvement. The following are the specific recommendations along with our responses:

Reviewer 1

In this study, authors measured levels of Aβ42, C-reactive protein (PCR) and TNFα in saliva from healthy subjects and patients with cancer, AD and coronary heart disease. However, no data, including demographic and clinical information, has been reported, thus it is impossible to compare any results, neither to get any conclusion.

Methods should be written more accurately. Demographic and clinical data of subject/patient groups is missing. Additionally, no data regards saliva collection is reported. Procedure of CRP assay is missing (sample used, volumen,...)

We added detailed demographic and clinical information of saliva donors in the supplemental tables 1-3.

P5-6: Material and methods section:

             Saliva (5-10 ml) from normal healthy controls and patients was collected in vials including 100 mL of thioflavin S and sodium azide (0.5 mg/ml each) to prevent amyloid aggregation induced by Abeta42 proteins and bacterial growth. Detailed demographic information of all the saliva donors (both normal controls and all the diseased cases) are provided in supplemental tables 1-3.

We added detailed procedures for CRP assay in the materials and methods section.

P7: Material and methods section:

For determining the C-reactive protein levels (CRPs), the ELISA protocol was utilised. The first step was adding a CRP capture antibody (mouse monoclonal, 1/2,000) to 96 microwell plates. The capture antibody was kindly supplied by Dr. Nijmeije (Amsterdam, the Netherlands). The plate was left overnight in a cold room (at 4°C) to allow binding of the antibody to. The remaining supernatants were then discarded. Bovine serum albumin (5%, 400 μL) in PBS-T was added to the plates to block non-specific binding to the capture antibody. The solution was then incubated at 20°C for 1 h. That BSA solution was discarded and the wells were washed twice with PBS.

Human saliva (100 μL) samples and CRP at various concentrations (0 - 50 ng/ml) were added to the plates and incubated at RT for 3 h. The saliva supernatants were then discarded. The wells were washed twice with PBS-T and were incubated with rabbit polyclonal anti-CRP antibody (100 μL, 1/2,000, Dako, Cat#: A073) at RT for 1.5 h. After that, the plate was washed twice with PBS-T HRP-conjugated polyclonal anti rabbit IgG antitibody (100 μL, 1/2,000, Dako, Cat#: P0448, DAKO, Mississauga, Ontario) for 1 h. The antibody was removed and the wells were washed twice with PBS-T and once with PBS.

For detection, a tetramethylbenzidine (TMB) solution (100 μL, Thermofisher Scientific, Cat#: N301) was added and the incubation carried out at 37°C for 5 min (The TMB solution was kept at 4°C until it was added for the color reaction). A stop solution of 100 μL (2 M sulfuric acid, Thermofisher Scientific, Cat#: N600) was then added to the well plates. The optical density was measured at 450 nm. The CRP concentration was calculated from a standard curve which was made with recombinant CRP (BioLegend, San Diego, CA).

Statistical analysis is completely deficiency. I recommend a complete revision for the statistical analysis: Use statistical tests accordingly (parametric or non-parametric). If ANOVA is used, which post hoc test is used to determine individual differences? For data visualization please use SD instead of SEM.

We added statistical analysis in the materials and methods section as the review required. We also replaced SEM with SD in the result section and figure legends.

P8: Material and methods section:

The significance of differences between data sets was analyzed by one-way ANOVA. Multiple group comparisons were followed by a post-hoc Bonferroni t-test. P values are given in figure legends.

Please rewrite Results to highlight the findings of this study.

We rewrote results section.

P9-12: Results section

We firstly investigated whether Abeta42 protein accumulation occurs in the brains of patients who died with post-mortem evidence of AD brains. A suitable control brain was available from a person died suddenly from a heart attack. The brains were stained with monoclonal Abeta42 antibody. Figure 1A and B demonstrate that Abeta42 accumulation in the hippocampal area is clearly evidenced in AD (Figure. 1A), but not in the healthy control (Figure 1B).

Since it is known that there is an Abeta42 increase in cerebrospinal fluid from AD patients, we examined whether there was a similar increase in their saliva. Saliva (5 - 10 ml) was collected in plastic vials. which also included 100 mL of thioflavin S and sodium azide (0.5 mg/ml each) to prevented not only aggregation induced by Abeta42 protein but also bacterial growth. Abeta42 levels were measured by an ELISA method. Protein amounts were analyzed by a standard curve made with commercial Abeta42 proteins. The data in Figure 2A show that in saliva the low level control group expressed protein levels that were remarkably constant between ages of 16 to 92, with no difference occurring between males and females (supplemental table 1). This low level control group expressed Abeta protein levels less than 30 pg/ml. The 148 low level control group, including one case with Parkinson disease (diagnosed 3 years previously), showed a mean ± SD of 21.26 ± 1.73 pg/ml. We also found that the 89 cases in the high level control group showed Abeta42 protein levels greater than the 148 low control group. This high level control group expressed levels greater than 30 pg/ml. They are believed to be at high risk for AD because their family history showed a 1.8 times higher Abeta42 levels than low controls (mean ± SD: 37.96 ± 8.13 pg/ml, P<0.01 compared with low level control group). The ages of this group were mainly distributed between 40 and 80 (Figure 2A).

We studied whether administration of ibuprofen (200 mg, twice a day) could reduce their Abeta42 protein levels. For this study eleven cases were selected in the high healthy control group. They orally administered ibuprofen (200 mg, twice a day) and measured their saliva Abeta42 protein levels in every one or two weeks. We found that all the high controls showed their salivary Abeta42 protein levels in the saliva to be reduced to low control levels within a month. Two of them are shown in Figure 2B.

We also measured salivary Abeta42 protein levels from patients with AD. The 30 AD cases, had been diagonised between 3-10 years previously in local clinics and hospitals. They showed Abeta42 protein levels even greater than the high level control groups (mean ± SD: 51.70 ± 10.50 pg/ml, P<0.01 compared with the high level control group). As a further control, we measured Abeta42 protein over throughout the day and found that there was no change in the protein levels (Figure 2C). Overall, these data demonstrate that measuring salivary Abeta42 levels can diagnose AD and indicate as well that it may predict the risk of future onset.

The next example we investigated was C-reactive proteins which is known to be involved in coronary heart disease. We examined the expression of the proteins in heart tissues. We immunostained the heart tissue from patients with coronary heart attack with sheep polyclonal anti-CRP antibody.   We found that expression of CRP was increased in heart tissue from patients died with coronary heart attack (Figure 1D) compared with tissue from a healthy control, which was from a patients died with a car accident (Figure 1C).

We used an ELISA assay method to measure salivary CRP protein levels. Data were analyzed with a standard curve with commercially available recombinant human CRP proteins. Figure 3 demonstrates the diagnosis and predictability of inflammation, especially coronary heart attack, by measurements of salivary CRP levels in 66 individuals. Healthy people show CRP levels of less than 3 ng/ml in their saliva (low level control group, 31 cases, mean ± SD: 1.67 ± 0.21). As shown for Abeta42, healthy controls (low level controls) produced CRPs constantly over ages from 12 to 90 with no gender difference. Diseases such as Parkinson’s disease and type II diabetes did not affect CRP expression (1.83 and 1.84 ng/ml, respectively).

However, high level control groups, which demonstrate CRP levels higher than 3 ng/ml, showed increased levels of CRP with no overlap with the low level control group (mean ± SD: 5.34 ± 1.70, P<0.01 compared with low level control group). This group is believed to be at risk for onset of inflammatory diseases, including heart disease. Demographically, everybody in this group showed a clear family history of cardiovascular disease such as heart attack or hypertension (data not shown).

The diseased group in both genders showed CRP levels high enough to suffer various diseases such as heart attack, transient ischemic attack (TIA), peripheral nerve failure and giant cell arthritis (mean ± SD: 15.20 ± 7.21, P<0.01 compared with the high level control group).

             Another example was Tumor-necrosis factor-a (TNF). For immunostaining we used a healthy control brain tissue from a person who died from a car accident. The brain tissue from a diseased case was from a patient who died from leukemia. We immunistained the capillary region of the brains with the rabbit polyclonal anti-TNF antibody. The data in Figure 1E and F show immunohistochemical staining for TNF in brain tissue from the healthy case (E), and the cancer case (F). Intensely immunostained capillaries in the cancer case compared to the normal case were found.

In order to confirm the results, we examined salival TNF protein levels. It was found that the predictability of cancer onset by measuring salivary TNF levels in 42 people is possible (Figure 4). Salivary TNF levels in 25 healthy volunteers, shown in supplemental table 3, ranging from 12 to 79 years of age were in the very narrow range of 155-175 pg/ml (control level group, mean ± SD: 159.18 ± 7.38). They are consistent with the presumed role of TNF in aiding capillaries to provide a normal blood supply to tissues. Individuals with various types of cancer (15 cases), diagnosed in local clinics and hospitals from 1-3 years previously, showed salivary TNF protein levels which were 2-3 fold higher than healthy control group. The values ranged from 260 pg/ml to 474 pg/ml (mean ± SD: 351.53 ± 63.8). Presumably this was because the malignancies demanded neo-capillarization to supply their needs. There was no overlap between the levels in normal and cancer cases (P<0.01 compared with control level group).

Page 3: "The data in Figure 2A ...expressed levels that were...". Levels of what? Where? In which sample? "...showed a mean of 21.26±0.14pg/ml". Mean of what? Where? These are only a couple of examples in the manuscript, but there are many others. These statements do not support the findings proposed by the authors in this manuscript.

We rewrote the part as reviewer requested.

P9: Results section

Since it is known that there is an Abeta42 increase in cerebrospinal fluid from AD patients, we examined whether there was a similar increase in their saliva. Saliva (5 - 10 ml) was collected in plastic vials. which also included 100 mL of thioflavin S and sodium azide (0.5 mg/ml each) to prevented not only aggregation induced by Abeta42 protein but also bacterial growth. Abeta42 levels were measured by an ELISA method. Protein amounts were analyzed by a standard curve made with commercial Abeta42 proteins. The data in Figure 2A show that in saliva the low level control group expressed protein levels that were remarkably constant between ages of 16 to 92, with no difference occurring between males and females (supplemental table 1). This low level control group expressed Abeta protein levels less than 30 pg/ml. The 148 low level control group, including one case with Parkinson disease (diagnosed 3 years previously), showed a mean ± SD of 21.26 ± 1.73 pg/ml. We also found that the 89 cases in the high level control group showed Abeta42 protein levels greater than the 148 low control group. This high level control group expressed levels greater than 30 pg/ml. They are believed to be at high risk for AD because their family history showed a 1.8 times higher Abeta42 levels than low controls (mean ± SD: 37.96 ± 8.13 pg/ml, P<0.01 compared with low level control group). The ages of this group were mainly distributed between 40 and 80 (Figure 2A).

What is "high controls"?

We added this in the text; high level control group: higher than 30 pg/ml in AbetaA42 levels in saliva.

P9: Results section

We also found that the 89 cases in the high level control group showed Abeta42 protein levels greater than the 148 low control group. This high level control group expressed levels greater than 30 pg/ml. They are believed to be at high risk for AD because their family history showed a 1.8 times higher Abeta42 levels than low controls (mean ± SD: 37.96 ± 8.13 pg/ml, P<0.01 compared with low level control group). The ages of this group were mainly distributed between 40 and 80 (Figure 2A).

Authors should improve the quality of immunohistochemistry images.

We improved the image from 300 dpi to 600 dpi.

Reviewer 2 Report

Here the authors have studied three selected biomarkers from saliva, as well as from heart (CRP) and elderly brain (Abeta42, TNF) tissues. The review-type title promises a lot – but instead of “to summarize current applications of saliva analysis and to point out promising directions for the future”, the paper in reality gives only a very limited view on this topic.

Introduction is poorly written and the fragmentary statements on selected markers hardly introduce the story to readers. Citations are also needed for statements like presented in lines 30 and 36. On the basis of the sentence “…but saliva…has largely been ignored” with a reference #5, it seems that the authors are unaware of a huge number of saliva studies published after year 1992. The aim of the study is missing!

The Mat&Met section presents the methods but the material part is completely missing. Where did the different types of samples (saliva, heart and brain tissues) come from? AD patients/non-AD individuals? CHD patients? Type II diabetics? Individuals with cancers? One patient with Parkinson disease? Who were those medicated with ibuprofen? How were the samples collected? Who were the brain tissue donors/heart tissue donors? Each sample type: how many samples? What are healthy people/normal cases/low controls/high controls?

The Results section is not understandable due to the missing description of study materials.

Conclusions drawn (abstract, the last discussion pagraph) are without a solid base supporting the own results presented in the paper.

The paper is confusingly written, which makes it really challenging to follow authors’ line of reasoning. There are fuzzy sentences throughout the manuscript text, and it is definitely needed to clarify what is meant (e.g. sentences in lines 31, 36-37, 44-45, 46-47, 176-177, 179-180, 184-185, 200-202, …). Therefore, a compact intro section, including a concise aim instead of the current one, as well as a clear presentation of the findings are needed for introducing and explaining the significance of the author’s work to readers.

Some minor comments: - fatal diseased cases? - TNFs/TNF-alpha/TNF: what have the authors precisely studied? - abbreviations should not be used before opened at the first time: AD, MW, RT… - all refs in the reference list need to be presented according to author instructions (one style)

Author Response

We are pleased to resubmit our manuscript "Saliva Diagnosis as a Disease Predictor (Manuscript ID: jcm-681440)" with revisions as suggested by the reviewers. We thank the reviewers for going over the manuscript so carefully and for making constructive suggestions for improvement. The following are the specific recommendations along with our responses:

Reviewer 2

Here the authors have studied three selected biomarkers from saliva, as well as from heart (CRP) and elderly brain (Abeta42, TNF) tissues. The review-type title promises a lot – but instead of “to summarize current applications of saliva analysis and to point out promising directions for the future”, the paper in reality gives only a very limited view on this topic.

We changed the title of the paper “Saliva Diagnosis as a Disease Predictor”

Introduction is poorly written and the fragmentary statements on selected markers hardly introduce the story to readers. Citations are also needed for statements like presented in lines 30 and 36. On the basis of the sentence “…but saliva…has largely been ignored” with a reference #5, it seems that the authors are unaware of a huge number of saliva studies published after year 1992. The aim of the study is missing!

We have rewritten the introduction section (adding references, the aim of studies and changed the sentence).

P3-4: Introduction

Saliva is a product of the salivary glands. It is expressed at a constant rate around the clock every day of life as is the case for all proteins generated from maintenance genes. Such genes do not express themselves here and there, and now and then as many have believed. They are expressed everywhere in the body at a rate specific for the tissue involved. Once the relative rates for tissues are known, they will apply to the proteins which are the products of their genetic activity [1]

A suitable example is amyloid beta protein terminating at position 42 (Abeta42). Immunohistochemical studies of brain tissue in Alzheimer disease (AD) cases have established intense staining for Abeta42 in affected regions. It was known that Abeta42 accumulation in AD is initiated by Abeta42 binding to the complement protein C1q, which initiate complement cascade resulting in neuronal death in affected area by membrane attack complex formation [2], which eventually results in the cognitive deficits that define clinical AD. In the affected areas the gene products have not been clearely removed. This means that differential effect of gene activity results in an integral effect of tissue accumulation. It has been suggested that this sequence of events can be avoided by early intervention with anti-inflammatory agents. Traditional non-steroidal anti-inflammatory drugs (NSAIDs) are one such class of agents. [3]. Therefore it is useful to investigate whether Abeta42 protein expression is increased in AD patients and whether oral administration of a NSAID such as ibuprofen attenuates its expression.

C-reactive proteins (CRPs) are an acute-phase protein produced in the liver and has been considered as a biomarker with plasma levels increasing in response to inflammation [4]. It was suggested that increased CRPs levels were found in cardiovascular diseases such as coronary heart disease [5]. Therefore a study of CRP levels in saliva and prediction of heart disease onset is a good example of relationship between gene activity and diseases.

            Tumor-necrosis factor-a (TNF) is produced as a 233 amino acid transmembrane protein of molecular weight of 26 kDa and stable in trimers [6]. It is known to be upregulated in inflammatory conditions such as malignant tumor expansion since it results in neo-capillary formation; cancers may be relatively benign until factors, yet to be identified, induce TNF production. This new, excessive production of TNF leads to a cachectic state. The apparent reason is that the proliferating cancer cells need TNF to stimulate their necessary blood supply. This requires new capillary formation. Complement activation is required to produce the necessary vascular endothelial growth factor for this to occur [6-7]. In order to clarify the functions of TNF, measuring salivary TNF levels as an indicator of its in vivo expression in healthy and cancerous groups will be necessary to understand relationship between TNF gene activity and cancer cell growth.

Urine and blood analyses have been routine for many decades but saliva, another body fluid, has largely less attended [8]. And yet it is beginning to appear that saliva may not only offer an index to the existence of a number of diseases but may even offer a clue to vulnerability to some where protective action can be taken.

The Mat&Met section presents the methods but the material part is completely missing. Where did the different types of samples (saliva, heart and brain tissues) come from? AD patients/non-AD individuals? CHD patients? Type II diabetics? Individuals with cancers? One patient with Parkinson disease? Who were those medicated with ibuprofen? How were the samples collected? Who were the brain tissue donors/heart tissue donors? Each sample type: how many samples? What are healthy people/normal cases/low controls/high controls?

We added detailed information of saliva donors in the supplemental tables 1-3. Heart and brain donors in results section. Other all the information was added in the results section.

P5-6 Materials and Methods section

Saliva (5-10 ml) from normal healthy controls and patients was collected in vials including 100 mL of thioflavin S and sodium azide (0.5 mg/ml each) to prevent amyloid aggregation induced by Abeta42 proteins and bacterial growth. Detailed demographic information of all the saliva donors (both normal controls and all the diseased cases) are provided in supplemental tables 1-3.

P9-12: results section

We firstly investigated whether Abeta42 protein accumulation occurs in the brains of patients who died with post-mortem evidence of AD brains. A suitable control brain was available from a person died suddenly from a heart attack. The brains were stained with monoclonal Abeta42 antibody. Figure 1A and B demonstrate that Abeta42 accumulation in the hippocampal area is clearly evidenced in AD (Figure. 1A), but not in the healthy control (Figure 1B).

Since it is known that there is an Abeta42 increase in cerebrospinal fluid from AD patients, we examined whether there was a similar increase in their saliva. Saliva (5 - 10 ml) was collected in plastic vials. which also included 100 mL of thioflavin S and sodium azide (0.5 mg/ml each) to prevented not only aggregation induced by Abeta42 protein but also bacterial growth. Abeta42 levels were measured by an ELISA method. Protein amounts were analyzed by a standard curve made with commercial Abeta42 proteins. The data in Figure 2A show that in saliva the low level control group expressed protein levels that were remarkably constant between ages of 16 to 92, with no difference occurring between males and females (supplemental table 1). This low level control group expressed Abeta protein levels less than 30 pg/ml. The 148 low level control group, including one case with Parkinson disease (diagnosed 3 years previously), showed a mean ± SD of 21.26 ± 1.73 pg/ml. We also found that the 89 cases in the high level control group showed Abeta42 protein levels greater than the 148 low control group. This high level control group expressed levels greater than 30 pg/ml. They are believed to be at high risk for AD because their family history showed a 1.8 times higher Abeta42 levels than low controls (mean ± SD: 37.96 ± 8.13 pg/ml, P<0.01 compared with low level control group). The ages of this group were mainly distributed between 40 and 80 (Figure 2A).

We studied whether administration of ibuprofen (200 mg, twice a day) could reduce their Abeta42 protein levels. For this study eleven cases were selected in the high healthy control group. They orally administered ibuprofen (200 mg, twice a day) and measured their saliva Abeta42 protein levels in every one or two weeks. We found that all the high controls showed their salivary Abeta42 protein levels in the saliva to be reduced to low control levels within a month. Two of them are shown in Figure 2B.

We also measured salivary Abeta42 protein levels from patients with AD. The 30 AD cases, had been diagonised between 3-10 years previously in local clinics and hospitals. They showed Abeta42 protein levels even greater than the high level control groups (mean ± SD: 51.70 ± 10.50 pg/ml, P<0.01 compared with the high level control group). As a further control, we measured Abeta42 protein over throughout the day and found that there was no change in the protein levels (Figure 2C). Overall, these data demonstrate that measuring salivary Abeta42 levels can diagnose AD and indicate as well that it may predict the risk of future onset.

The next example we investigated was C-reactive proteins which is known to be involved in coronary heart disease. We examined the expression of the proteins in heart tissues. We immunostained the heart tissue from patients with coronary heart attack with sheep polyclonal anti-CRP antibody.   We found that expression of CRP was increased in heart tissue from patients died with coronary heart attack (Figure 1D) compared with tissue from a healthy control, which was from a patients died with a car accident (Figure 1C).

We used an ELISA assay method to measure salivary CRP protein levels. Data were analyzed with a standard curve with commercially available recombinant human CRP proteins. Figure 3 demonstrates the diagnosis and predictability of inflammation, especially coronary heart attack, by measurements of salivary CRP levels in 66 individuals. Healthy people show CRP levels of less than 3 ng/ml in their saliva (low level control group, 31 cases, mean ± SD: 1.67 ± 0.21). As shown for Abeta42, healthy controls (low level controls) produced CRPs constantly over ages from 12 to 90 with no gender difference. Diseases such as Parkinson’s disease and type II diabetes did not affect CRP expression (1.83 and 1.84 ng/ml, respectively).

However, high level control groups, which demonstrate CRP levels higher than 3 ng/ml, showed increased levels of CRP with no overlap with the low level control group (mean ± SD: 5.34 ± 1.70, P<0.01 compared with low level control group). This group is believed to be at risk for onset of inflammatory diseases, including heart disease. Demographically, everybody in this group showed a clear family history of cardiovascular disease such as heart attack or hypertension (data not shown).

The diseased group in both genders showed CRP levels high enough to suffer various diseases such as heart attack, transient ischemic attack (TIA), peripheral nerve failure and giant cell arthritis (mean ± SD: 15.20 ± 7.21, P<0.01 compared with the high level control group).

             Another example was Tumor-necrosis factor-a (TNF). For immunostaining we used a healthy control brain tissue from a person who died from a car accident. The brain tissue from a diseased case was from a patient who died from leukemia. We immunistained the capillary region of the brains with the rabbit polyclonal anti-TNF antibody. The data in Figure 1E and F show immunohistochemical staining for TNF in brain tissue from the healthy case (E), and the cancer case (F). Intensely immunostained capillaries in the cancer case compared to the normal case were found.

In order to confirm the results, we examined salival TNF protein levels. It was found that the predictability of cancer onset by measuring salivary TNF levels in 42 people is possible (Figure 4). Salivary TNF levels in 25 healthy volunteers, shown in supplemental table 3, ranging from 12 to 79 years of age were in the very narrow range of 155-175 pg/ml (control level group, mean ± SD: 159.18 ± 7.38). They are consistent with the presumed role of TNF in aiding capillaries to provide a normal blood supply to tissues. Individuals with various types of cancer (15 cases), diagnosed in local clinics and hospitals from 1-3 years previously, showed salivary TNF protein levels which were 2-3 fold higher than healthy control group. The values ranged from 260 pg/ml to 474 pg/ml (mean ± SD: 351.53 ± 63.8). Presumably this was because the malignancies demanded neo-capillarization to supply their needs. There was no overlap between the levels in normal and cancer cases (P<0.01 compared with control level group).

The Results section is not understandable due to the missing description of study materials.

We added all the study materials and detailed information in the results section. Please see above.

Conclusions drawn (abstract, the last discussion paragraph) are without a solid base supporting the own results presented in the paper.

We changed conclusion part in the abstract and discussion part.

The paper is confusingly written, which makes it really challenging to follow authors’ line of reasoning. There are fuzzy sentences throughout the manuscript text, and it is definitely needed to clarify what is meant (e.g. sentences in lines 31, 36-37, 44-45, 46-47, 176-177, 179-180, 184-185, 200-202, …). Therefore, a compact intro section, including a concise aim instead of the current one, as well as a clear presentation of the findings are needed for introducing and explaining the significance of the author’s work to readers.

Most of this paper has been rewritten and now understandable.

Some minor comments: - fatal diseased cases? - TNFs/TNF-alpha/TNF: what have the authors precisely studied? - abbreviations should not be used before opened at the first time: AD, MW, RT… - all refs in the reference list need to be presented according to author instructions (one style).

Abbreviations and reference section was changed as the reviewer requested.

Round 2

Reviewer 1 Report

Authors have addressed point-by-point the reviewer comments, they have provided the new data/experiments/re-analysis, and they have included them in the revised manuscript which considerably strengthen the role of saliva as promissing diagnostic tool.